# Influence of Prescribed Burning on a *Pinus nigra* subsp. *Laricio* Forest: Heat Transfer and Tree Vitality

**Lila Ferrat \*, Frédéric Morandini** 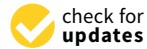 **and Gauthier Lapa**

UMR CNRS 6134, Campus Grimaldi, University of Corsica, 20250 Corte, France;
morandini_f@univ-corse.fr (F.M.); lapa_g@univ-corse.fr (G.L.)
**\*** Correspondence: ferrat_l@univ-corse.fr

**Abstract:** Surface fuel removal is crucial to facilitate the mitigation of severe fires in forests. Prescribed burning is often used by forest managers, thanks to its low cost and high efficiency in hard-to-reach areas. The determination of heat transfer between fires and trees has rarely been carried out on living species and consequently, their long-term effects on tree physiology are still not fully understood. In this study, a multidisciplinary approach was conducted to evaluate the impact of a late spring (June) prescribed burning on a Mediterranean pine forest (*Pinus nigra* subsp. *laricio*). The surface fuels consisted of a $656 \, \text{g m}^{-2}$ needle litter, mixed with a few scattered living herbaceous strata. During the fire spread, measurements of the inner and outer trunk temperatures were made at the base of 12 trees with an average bark thickness of $19.4 \pm 7.0$ mm. The fireline intensity and flame residence time were in the range of $110–160 \, \text{kW m}^{-1}$ and $220–468$ s, respectively. Despite a maximum heating rate at the cambial area of $4.37 \, ^\circ\text{C min}^{-1}$, the temperature of these tissues remained below $60 \, ^\circ\text{C}$, a critical threshold above which thermal damage will occur. In addition, prior- and post-fire physiological monitoring was performed over a long time period (2.5 years) on 24 trees, using sap flow, chlorophyll fluorescence and gas exchange measurements. All parameters remain highly correlated and indicate that the burned trees did not suffer physiological damage. Moreover, drought resistance strategies were not altered by the prescribed burning. The thermal insulation capability of the bark allowed the functional tissues to experience low heat stress that did not affect tree vitality.

**Keywords:** prescribed burning; *Pinus nigra* subsp. *laricio*; sap flow; chlorophyll fluorescence; gas exchange; thermal measurements

## 1. Introduction

Wildfires represent a significant source of disturbance in Mediterranean ecosystems, impacting both economic and social activities [1]. After nearly a century of fire exclusion across forest ecosystems and the reduction of agricultural practices, surface and canopy fuels have accumulated. The horizontal and vertical continuity of fuels in the landscape has triggered large and high-severity fires [2,3]. Given 21st-century climate change scenarios that will increase this trend [4,5], fire managers face huge challenges in terms of risk and land management. Pines, playing a prominent role in Mediterranean forests due to their widespread distribution, are particularly affected by wildfires. They account for a large proportion (>2/3) of the total burned areas [1,2].

Although the implementation of better forest fire prevention policies has considerably reduced the number of hectares burned in recent decades, more than 600,000 ha of vegetation are still reduced to ashes each year in the Mediterranean area [6]. Means of prevention against forest fires have been strengthened. They include increased public information and raising awareness about wildfire risk, but first and foremost, they aim at reducing fire hazards. This is achieved by installing and maintaining natural areas in which the fuel load at ground level is low. Indeed, the excessive accumulation of hazardous understory flammable materials can be rapid under pine trees (needles, wooden debris, shrubs).

Their removal can be performed using mechanical thinning and/or reintroducing fire into ecosystems through prescribed burning [7–10]. The latter is time-saving and allows to circumvent logistical and cost constraints in difficult-to-reach areas and on steeply sloping terrain. The removal of these ladder fuels will reduce the potential of fire transition from the surface to the crown and create more spatial variability in the structure of the forest [11]. Wildfire risk and severity are thus greatly mitigated for the next fire-prone seasons [12,13].

Despite the emphasis on the benefits of fire restoration in ecosystems in terms of fire prevention and ecological benefits (creation of a mosaic of vegetation types and habitats, increased floristic diversity [3,14,15], prescribed burning remains a controversial approach worldwide [16–18]. This forest management practice is often accused of causing pollution (particles and carbon dioxide release, water quality after rainfall), reducing biodiversity and ecosystem disturbances [19–24].

Several studies have been conducted on quantifying the effects of fire-related heat stress on trees. These works have mainly focused on the morphological and physiological damage to the trunk or crown (Table 1).

In addition, some authors have also built statistical models for the prediction of tree mortality, based on post-fire observations of these fire impacts, such as bark charring and crown scorch [25–28]. Obviously, the base of the trunks is particularly prone to heat impingement, since it is in direct contact with the flames from a surface fire. Previous research has focused on tree-ring and fire scar analysis [29–32] and underline the important role of the bark in protecting the functional tissues (cambium, phloem and xylem) during a surface fire or prescribed burning [33–36]. As a result, the base can be exposed to high temperatures over a prolonged period of time, depending on the fuel load at ground level.

The rate of heat transmission to vascular tissues increases with decreasing bark thickness and inner bark moisture content [36]. Numerous studies focusing on heat transfer into the trunk and the resulting damage mainly used artificial heating methods that allow the controlled and localized heating of the trunk. Cotton rope soaked in fuel [37,38], a gas torch [39,40], a radiant heat source [41] or a hot water bath [42,43] were used to study the effects of heat stress [37,44–49] or the depth of tissue necrosis [38,47] cell mortality [42] and the cavitation and deformation of xylem tissues [43,50].

**Table 1.** Synthesis of experimental research on the effects of heat on the trunk and crown.

| Location | Species | Heating Method | Key Words | Authors [ref.] |
|---|---|---|---|---|
| **Stem** | | | | |
| | *Pinus strobus* | gas torch | cambium temperature, heat conduction | Kayll [44] |
| | *longleaf pines* | litter fires: headfires and backfires | leeward and windward trunk sides, lethal cambium temperatures | Fahnestock and Hare [51] |
| | *Ecclinusa* sp., *Inga* sp., *Jacaranda copaia*, *Pourouma guianensis*, *Macrolobium angustifolium*, *Diospyros duckei*, *Tetragastris altissima*, *Inga alba*, *Metrodorea flavida*, *Xylopia aromatica*, *Cecropia sciadophylla*, *Cordia sericalyx*, *Lecythis idatimon*, *Lecythis lurida*, *Manilkara huberi* | kerosene-soaked cotton rope | cambium temperature, heat conduction | Uhl and Kauffman [45] |
| | *Pinus pinaster* | straw | temperature distribution | Costa et al. [52] |
| | *Sequoiadendron giganteum*, *Pinus lambertiana* | natural litter | cambium temperature | Sackett and Haase [53] |
| | *Pinus ponderosa* | litter, prescribed burning | growth, dendroecology, isotopic | Peterson et al. [29] |
| | *Pinus halepensis* | electrical heating strip | physiological and growth responses | Ducrey et al. [54] |
| | Laboratory model | flame burner | fire scars, vortices | Gutsell and Johnson [55] |

**Table 1.** *Cont.*

| Location | Species | Heating Method | Key Words | Authors [ref.] |
|---|---|---|---|---|
| | *Anadenanthera colubrina, Poeppigia procera, Peltogyne heterophylla, Phyllostylon rhamnoides, Caesalpinia floribunda, Aspidosperma rigidum, Chorisia speciosa, Acacia polyphylla, Tabebuia impetiginosa, Centrolobium microchaete, Eriotheca roseorum, Machaerium scleroxylon, Astronium urundeuva, Spondias mombin, Ceiba samauma, Amburana cearensis* | kerosene-soaked cotton rope | cambium temperature, heat conduction | Pinard and Huffman [46] |
| | *Anadenanthera macrocarpa, Aspidosperma macrocarpon, Astronium urundeuva, Centrolobium microchaete, Machaerium scleroxylon, Poeppigia procera* | gas torch | stem wounding | Schoonenberg et al. [39] |
| | *Pinus ponderosa, Pseudotsuga menziesii, Abies concolor, Calocedrus decurrens* | electrical heating pad | cambium temperature, heat conduction | Van Mantgem and Schwartz [41] |
| | *Pinus contorta, Populus tremuloides, Picea engelmannii, Pseudotsuga menziesii* | water bath | cambium necrosis | Dickinson and Johnson [42] |
| | *Pseudotsuga menziesii* | kerosene-soaked cotton rope | heat conduction | Jones et al. [37] |
| | *Pseudotsuga menziesii, Pinus ponderosa, Acer rubrum, Quercus prinus* | oven, hot air, gas torch | bark thickness, modelization | Butler et al. [25] |
| | *Acer rubrum, Quercus prinus* | kerosene-soaked cotton rope or artificial fuel bed | fire behavior/bark thickness, tissue necrosis | Bova and Dickinson [47] |
| | *Acacia karroo* | gas torch | stem death, xylem conductivity | Balfour and Midgley [40] |
| | *Acer rubrum, Quercus prinus, Pinus ponderosa, Pseudotsuga menziesii* | kerosene-soaked cotton rope or artificial fuel bed | heat conduction, kill depth | Jones et al. [38] |
| | *Quercus rubrum, Fraxinus americana, Robinia pseudoacacia* | radiant heater | heat flux | Bova and Dickinson [56] |
| | *Fagus sylvatica, Abies alba, Tilia cordata, Pinus sylvestris, Larix decidua, Quercus suber, Sequoiadendron giganteum* | Bunsen burner | bark insulation capacity | Bauer et al. [33] |
| | *Pinus sylvestris* | wildfire | dendrochronology, isotopes | Beghin et al. [30] |
| | *Buchanania obovata, Callitris intratropica, Erythrophleum chlorostachys, Eucalyptus miniata, Corymbia polysciada, Eucalyptus tetrodonta, Terminalia ferdinandiana* | paraffin-soaked cotton rope | cambium temperature, heat conduction | Lawes et al. [48] |
| | *Populus balsamifera* | water bath | cavitation and deformation of xylem | Michaletz et al. [43] |
| | *Pinus halepensis* | litter, prescribed burning | dendrochronology, isotopes | Battipaglia et al. [31] |
| | *Pinus pinaster, Pinus radiata, Pinus elliottii, Eucalyptus cladocalyx, Acacia mearnsii, Ekebergia capensis, Rhus viminalis, Olea africana* | electric heat gun | bark thickness, moisture content, heat insulation capacity | Odhiambo et al. [34] |
| | *Pinus halepensis, Pinus nigra salzmanii, Pinus nigra* subsp. *nigra, Pinus sylvestris* | litter, prescribed burning | growth, dendrochronology | Valor et al. [32] |
| | *Pinus pinea* | litter, prescribed burning | hydraulic conductivity, radial growth | Battipaglia et al. [57] |
| | *Picea abies, Pinus sylvestris, Fagus sylvatica* | submersion in heated water | hydraulic conductivity, xylem | Bar et al. [50] |
| | *Pinus pinea* | mass loss calorimeter | bark thickness, flammability, cambium | Madrigal et al. [36] |
| | *Pinus nigra* | mass loss calorimeter device in a vertical configuration, low-intensity prescribed burning | bark thickness, cambial damage, fire intensity, time of heat exposure | Espinosa et al. [58] |
| **Crown** | | | | |
| | *Acer rubrum, Rubus allegheniensis, Prunus serotine, Quercus ellipsoidalis* | litter, prescribed burning | leaf nutrients, physiology, reproduction | Reich et al. [59] |
| | *Abies concolor, Pinus lambertiana, Pinus ponderosa, Calocedrus decurrens, Sequoiadendron giganteum, Quercus kelloggii* | litter, prescribed burning | scorch, tree mortality, models | Stephens and Finney [60] |
| | *Pinus pinaster* | litter, prescribed burning | crown scorch height, fire severity | Fernandes and Botelho [9] |

**Table 1.** *Cont.*

| Location | Species | Heating Method | Key Words | Authors [ref.] |
|---|---|---|---|---|
| | *Pinus ponderosa* | Litter, thinning, prescribed burning | leaf nutrients, photosynthesis, hydraulic conductance | Skov et al. [61] |
| | *Pinus nigra* subsp. *laricio* | litter, prescribed burning | secondary metabolites, isotopes | Cannac et al. [62] |
| | *Pinus nigra* subsp. *laricio* | litter, prescribed burning | secondary metabolites | Cannac et al. [63] |
| | *Pinus nigra* subsp. *laricio* | litter, artificial prescribed burning | secondary metabolites | Cannac et al. [64] |
| | *Pinus nigra* subsp. *laricio* | litter, artificial prescribed burning | physiology, biochemistry | Ferrat et al. [65] |
| | *Pinus palustris* | litter, prescribed burning | physiology | O'Brien et al. [66] |
| | *Pinus nigra* subsp. *laricio* | litter, prescribed burning | secondary metabolites | Cannac et al. [67] |
| | *Pinus halepensis, Pinus nigra* subsp. *laricio* | litter, prescribed burning | secondary metabolites, physiology | Lavoir et al. [68] |
| | *Pinus rigida* | litter, prescribed burning | physiology, isotopes | Renninger et al. [69] |
| | *Quercus mongolica* | butane torch | bark property, cambium, heat transfer, stem heating model | Wei et al. [70] |
| **Crown and Stem** | | | | |
| | *Pinus ponderosa* | litter, prescribed burning | physiology, morphology | Feeney et al. [71] |
| | *Pinus pinaster* | gas flame | pigments, tannins, polyphenols | Alonso et al. [72] |
| | *Pinus halepensis, Pinus pinea* | wildfire | tree mortality | Rigolot [73] |
| | *Pinus sylvestris, Pinus nigra, Pinus canariensis, Pinus radiata, Pinus pinaster, Pinus pinea, Pinus halepensis, Pinus brutia* | wildfire, prescribed burning | tree mortality, models, growth | Fernandes et al. [74] |
| | *Pinus nigra* subsp. *laricio* | litter, prescribed burning | secondary metabolites | Cannac et al. [75] |
| | *Pinus halepensis, Pinus pinaster, Pinus pinea* | wildfire | Tree mortality, scorch, modeling, bark thickness, | Pimont et al. [27] |
| | *Pinus pinaster* | radiation heat flux | physiology, cambium, growth | Jimenez et al. [49] |
| | *Pinus nigra, Pinus pinaster* | litter, prescribed burning | short-term effects, bark thickness, low fire intensity, cambium and bark temperatures, time of heat exposure, litterfall, LAI | Espinosa et al. [76] |
| | Synthesis | wildfire | modeling, biophysical processes | Butler and Dickinson [26] |
| | Synthesis | wildfire | bark thickness, fire regime | Pausas [35] |

However, even if temperature and exposure time were strongly controlled, the substituting processes for heating used in these previous studies are not representative of heat stress induced by a surface fire under field conditions. Studying fire under field conditions is very important to really understand its effects. During prescribed burning, the temperature and intensity of the fire are not constant in time and space, neither is the resulting heat transfer among all trees of the plot. The temperature increase at the base of a stem is also not homogeneous, particularly in the case of plants with rough and deeply fissured bark, like pines. Moreover, when a fire passes by a tree, due to turbulence, the development of vortices at the leeward side results in the presence of a standing flame. The persistence of this leeward flame increases the fire's residence time. Greater fire scars on the leeward side of the stem are a well-known occurrence [51,55]. Consequently, trunk temperature measurements performed during prescribed burning are a worthy topic for investigation. Furthermore, previous study [58] highlighted the importance of field data to confirm laboratory findings on heat exposure to samples of bark. Nevertheless, given the inherent challenge in physics-based measurements at field scale [77], few works have been conducted on living trees in this regard [38,49,53,76]. Many of them settle for air or bark surface measurements but the relationship between outer and inner bark temperature is not provided [51,57]. Other works used freshly cut stem segments, but boundary conditions and the absence of sap flow transport is likely to bias the representability of the measurements [33,37,38,52,70].

Combined with the monitoring of physiological indicators, the temperature measurements inside the trunk can greatly improve the understanding of the conduction heat transfer process through the bark into the vascular tissues. The accurate evaluation of these mechanisms should provide relevant insights on potential in-depth injuries to trunk tissues and their resulting effects on global tree physiology. In particular, the residence time above the lethal threshold of 60 °C is also a good predictive indicator of the risk of damage [57]. Alternatives to temperature measurements are temperature predictions using numerical approaches or empirical relationships [37,38,47,70,78–80]. Stem heating, tissue necrosis and resulting tree vitality or mortality can be predicted based on the fire properties, tree characteristics and injury thresholds. The critical residence time needed to inflict damage can be estimated based on the bark thickness and outer temperature [81]. Other approaches simulate the energy transfer process from the fire into the stems [26,38].

When a surface fire spreads, the significant buoyancy-driven convective heat transfer from hot combustion products may also impinge upon the tree's crown. At present, those studies that focus on branches and leaves are mainly based on leaf litter and undergrowth fires. These field studies use physiological [59,65,66,68,69] or chemical parameters as indicators of the effects of fire on trees (Table 1; [62–64,67–69]. Again, differences in tree species, burning practices (uphill or downhill fire spread) and environmental conditions (for example, in the cited literature, the litter load ranges from 250 to 1600 g m$^{-2}$) might cause significant variations in how tree vitality is impacted [9,60,68,74,82].

In previous works [72,73], attention was paid to parameters that allow linking impacts on trunk and crown, sometimes with respect to different combinations of heat treatment on both parts. Sap flow measurements were used to study water movement within the whole tree [83] and transpiration [84]. A decrease in sap flow may be explained in different ways, including the phenomenon of xylem cavitation, caused by thermal damage to the trunk [54], by a decrease in leaf area [69] and by perturbations of leaf gas exchange (e.g., maximum carboxylation rate (Vcmax), stomatal conductance ($g_s$) and photosynthesis (A) [59,69]. Noticed effects on growth or physiological performances after prescribed burning were also explained by modifications of soil nutrient concentration (see [82] for a review) and soil water availability [61,71] although both effects do not persist over time.

Few studies have been done on the resistance of *Pinus nigra* subsp. *laricio* (Poir.) Maire var. *corsicana* (Loudon) Hyl (*Pinus laricio*) to fire. It appears that its regeneration capacities are lower than those of species like *Pinus pinaster* and *Pinus halepensis*, the serotinous cones of which disperse seeds after fires [85] and which have adaptations that help perpetuation in post-fire regeneration areas [74]. Its forests, which characterize the Corsican mountains in the supra-Mediterranean and montane zones (900 to 1800 m), are regularly treated by understory burning for fire-risk and hazard reduction purposes. For a more durable and effective cleaning before the wildfire season, these burnings, usually carried out in autumn and winter, would need to be extended as late as possible in the spring. The concern of weakening trees by subjecting them to heat stress in periods of high metabolic activity [50] would be a constraint to this practice, especially in Mediterranean ecosystems regularly subjected to summer droughts [57,86,87].

The present study aims to investigate if a prescribed burning, realized beyond the limits recognized by forest agents, in terms of season, is capable of sufficiently disrupting the functioning of the trees to threaten their survival. To this intent, the intensity of heat transfer into the trunk during prescribed burning was measured and linked with physiological surveys (i.e., (a) sap flow, (b) chlorophyll fluorescence and (c) gas exchange) to assess tree vitality. Understanding the prescribed burning dynamic and its effects on tree ecophysiology is key to defining burn prescriptions [16].

## 2. Materials and Methods

### 2.1. Site Description

The study was carried out in a natural pure stand of *Pinus laricio* located in Corsica, France (42°11′48.5″ N, 9°05′39.6″ E) at 1200 m above sea level. This forest, managed by

the French Forest Office, has not been thinned nor harvested for more than 30 years. The understory consists of shrubs such as *Erica arborea* L., *Juniperus oxycedrus* L. and *Genista salzmanii* DC. The herbaceous layer is mostly represented by *Deschampsia flexuosa* L., *Brachypodium pinnatum* (L.) P. Beauv. and by a consistent needle litter layer (656 $\pm$ 28.3 g m$^{-2}$). The soil is a shallow-depth brown, with many rocky outcrops. The bedrock is granitic and the slope is about 70–80% with a northeast aspect. Meteorological trends are similar during the study period, with relatively hot, dry summers and mild winters. In 2014, the mean temperature was 10.2 °C and the total precipitation was 966 mm. The end of summer 2014 was particularly dry, with 41 rainless days (July 31–September 9) and only 14.5 mm of rainfall in September. Important daily and seasonal variations of vapor pressure deficit (VPD) were observed, with notably high values (>2 kPa) during the summer (Figure 1).

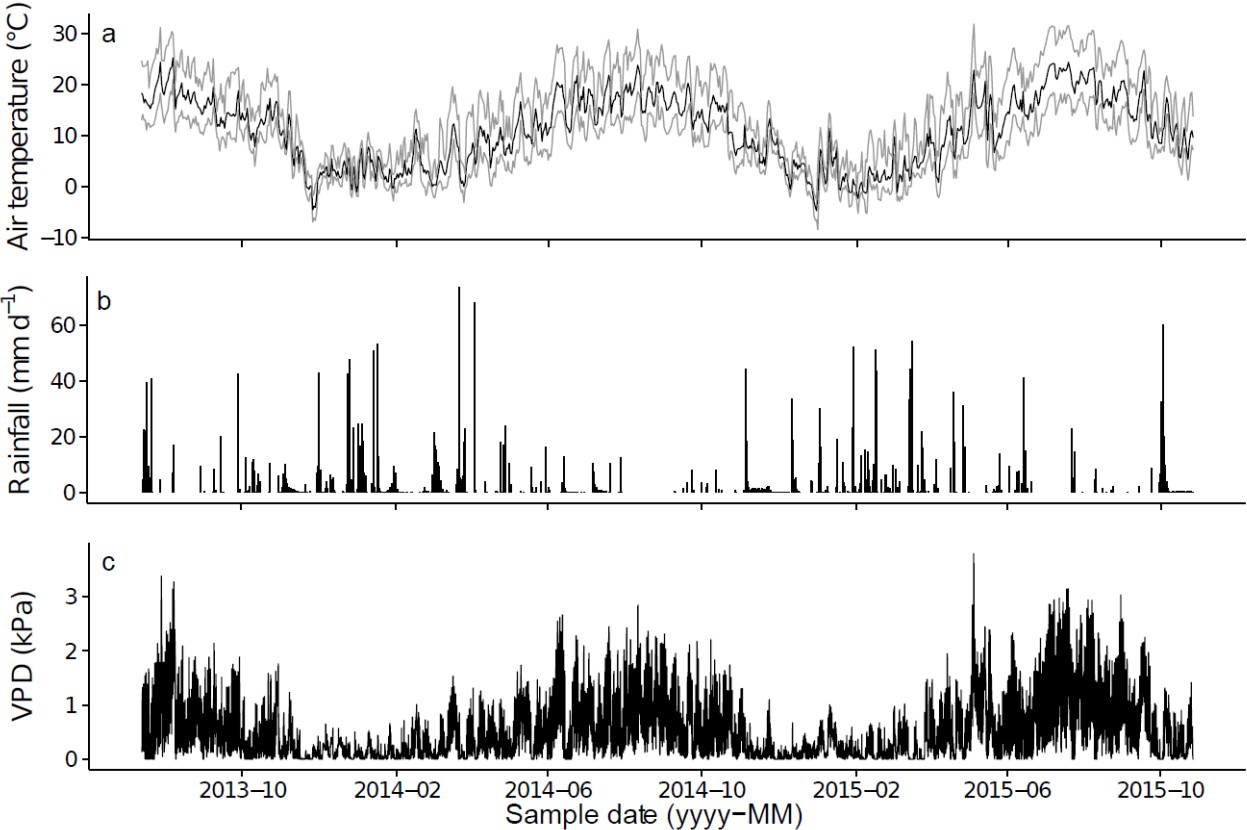

**Figure 1.** Daily (**a**) mean (black line), minimum and maximum (grey lines) air temperature (°C), (**b**) total rainfall (mm d$^{-1}$) and (**c**) continuous VPD (kPa), between 13 July 2013 and 26 October 2015. The dashed vertical line represents the date of the prescribed burning.

The stand was divided into a 300 m$^2$ control unburned plot and a 400 m$^2$ burned one. The basal area of the control and burned plots were 36.5 and 31.5 m$^2\cdot$ha$^{-1}$, respectively. It should be noted that the control plot was more than 50 m away and was shifted from the experimental one so as not to be affected by the fire and released smoke. The competition index for all pines was measured using Hegyi's method [88]. Twenty-four co-dominant pines were randomly selected for physiological measurements (diameter = 16.4 $\pm$ 3.4 cm; mean height = 10.2 $\pm$ 2 m; mean bark thickness at 20 cm above ground = 19 $\pm$ 4 mm; all values are mean $\pm$ SD; see Tables 2 and 3).

**Table 2.** Diameter at breast height (DBH), height and age of the control trees and length of the heat dissipation probes (TDP). Chlorophyll fluorescence and gas exchange measurements were made on the trees indicated by an asterisk.

|  | C1 * | C2 * | C3 * | C4 * | C5 | C6 * | C7 * | C8 * | C9 * | C10 | C11 * | C12 | Mean (SD) |
|---|---|---|---|---|---|---|---|---|---|---|---|---|---|
| DBH (cm) | 15.0 | 14.5 | 13.0 | 12.5 | 12.5 | 13.5 | 19.0 | 18.0 | 21.0 | 21.5 | 20.0 | 21.0 | 16.8 (3.6) |
| Height (m) | 11.8 | 13.1 | 8.6 | 6.9 | 7.3 | 11.5 | 8.6 | 8.6 | 10.1 | 8.0 | 10.1 | 12.7 | 9.8 (2.1) |
| Age (year) | 45 | 38 | 36 | 29 | 23 | 22 | 29 | 29 | 42 | 42 | 50 | 43 | 35.7 (9.1) |
| TDP (mm) | 30 | 30 | 30 | 30 | 30 | 30 | 50 | 50 | 50 | 50 | 50 | 50 | - |

**Table 3.** Diameter at breast height (DBH), height and age of the burned trees and length of heat dissipation probes (TDP). Chlorophyll fluorescence and gas exchange measurements were made on the trees indicated by an asterisk.

|  | B1 * | B2 * | B3 | B4 | B5 * | B6 * | B7 * | B8 * | B9 * | B10 * | B11 * | B12 | Mean (SD) |
|---|---|---|---|---|---|---|---|---|---|---|---|---|---|
| DBH (cm) | 16.0 | 16.0 | 12.0 | 12.5 | 14.0 | 13.0 | 17.0 | 17.0 | 20.8 | 17.0 | 23.0 | 19.3 | 16.5 (3.4) |
| Height (m) | 9.0 | 8.8 | 10.7 | 8.6 | 7.8 | 7.9 | 12.7 | 9.7 | 11.1 | 13.3 | 11.6 | 13.0 | 10.4 (2.0) |
| Age (year) | 28 | 33 | 43 | 24 | 31 | 29 | 41 | 49 | 45 | 35 | 45 | 42 | 37.1 (8.1) |
| TDP (mm) | 30 | 30 | 30 | 30 | 30 | 30 | 50 | 50 | 50 | 50 | 50 | 50 | - |

## 2.2. Prescribed Burning Instrumentation

The prescribed burning was carried out on 11 June 2014 between 10:30 and 12:30 by trained forest managers. The upper slope of the burn unit was lit by hand using drip torches. This intentionally created a backing fire with a low fire intensity. The fire spread was controlled on the edges of the plot using fire rakes. Backpack pumps were also used in some rare cases to prevent the initiation of fire transition toward the crown due to the presence of resin on the bark of the pine.

Previous studies on prescribed burning across pine needle beds exhibited strong temperature variations along the vertical, with maximum temperature occurring near the ground [64,75]. In the present work, temperature measurements were only focused on this particular high-temperature-prone region, located at the base of trees. Twelve trees were instrumented using thermocouples on the leeward side of the trunk (facing upslope). This side was chosen since greater injuries and scars occur on the bark due to flame vortices [52,89]. Two measurements were performed for each tree at 20 cm above the ground, both inside and outside the trunk. In order to place the thermocouples as near as possible to the cambium, 1-mm diameter holes were drilled over the bark nearly perpendicularly to the stem radius. The K-Type chromel-alumel thermocouples (Omega Engineering, Inc. Stamford, CT, USA) were inserted into the drilled holes under the bark. Thin junctions (250 μm shield with 50 μm wire diameter) were used for a fast response time (<0.4 s) and to minimize measurement errors (radiation and heat retention) within the flaming region. Grounded thermocouples are used because they are more rugged than exposed junctions, in particular for measurements inside the wood. The thermocouples body was insulated using multi-layer insulation materials based on ceramic to prevent measurement bias due to heat conduction along the stainless-steel sheath to the junction. Extension cables were buried underground to prevent thermal degradation during the fire. The whole set of thermocouples was plugged into 3 synchronized battery-powered data loggers (CR3000, Campbell Scientific Ltd., Loughborough, UK) located outside the burn area. In order to minimize the length of the extension cables (<10 m), three synchronized data loggers were used. Temperature measurements were recorded at a 1 Hz sampling rate. These measurements allowed identification of the maximum temperature and the residence time (above a determined threshold) for each of the locations studied (air and under bark). The fireline intensity (kW m$^{-1}$) was also derived from the Byram equation [90] as the product of the fuel consumed (kg m$^{-2}$), the fire rate of spread (m s$^{-1}$) and the heat generated by combustion (about 18,000 kJ kg$^{-1}$ for vegetal fuels).

Litter samples were taken before and after burning at five points for each plot, over an area of 0.25 m$^2$. The needles were dried at 60 °C for at least 48 h, up to the constant mass. Litter mass is expressed in g m$^{-2}$ of dry weight.

## 2.3. Meteorological Measurements

General meteorological measurements were provided by a weather station located in a clearing a few meters from the plot. Air temperature and relative humidity (CS215, Campbell Scientific Ltd., Loughborough, UK) were measured every 30 s and averaged over 30 min. Rainfall measurements (Rain Gauge 52203, R. M. Young Corporation, Traverse City, MI, USA) were collected every 30 min. Data were recorded using a battery-powered CR1000 data logger (Campbell Scientific Ltd., Loughborough, UK), recharged daily by a photovoltaic power source.

## 2.4. Physiological Measurements

Sap flow was measured on the 24 selected trees (12 control and 12 burned), between July 2013 and October 2015. Among them, 18 trees (9 control and 9 burned) were monitored for chlorophyll fluorescence and gas exchange, between December 2013 and July 2015. Sun-exposed branches, approximately five to six meters high, were sampled on the same orientation (northeast) and crown position. They were immediately re-cut under water and stored thus to avoid cavitation [91]. Preliminary experiments on *Pinus laricio* demonstrated that cutting did not affect the gas exchange or chlorophyll fluorescence up to 12 h after the cut. Similar results were observed with other conifers, including *Pseudotsuga menziesii* (Mirb.) Franco and *Pinus pinaster* Aiton [91,92]. To ensure the implementation of reliable measures and to avoid senescent needles, measurements were always carried out on current-year needles [92,93] at the end of the branch.

Sap flow was measured with thermal dissipation probes (TDP) according to Granier's method [94,95]. Either TDP-30 or TDP-50 (Dynamax Inc., Houston, TX, USA) were used, depending on the trunk diameter (Tables 2 and 3). Sensors were installed at breast height (1.30 m) with the same aspect (northeast) and protected with reflective insulation. Sap flow was calculated according to [95] and expressed as $dm^3 \, dm^{-2} \, h^{-1}$ or as daily total flow ($dm^3 \, dm^{-2} \, d^{-1}$). It was monitored between 13 July 2013 and 26 October 2015. Measurements were taken every 30 s and averaged every 30 min. Data were recorded into a battery-powered data logger (CR3000, Campbell Scientific Ltd., Loughborough, UK) recharged daily by solar panels. During winter, due to the low solar illumination in this mountain area, the recharging of a battery by solar panels was limited. Thus, only one tree was monitored for each plot (B1 and C2) between 15 November 2013 and 28 April 2014, and between 4 November 2014 and 24 April 2015. Other TDPs were turned off after the reduction and stabilization of sap flow in winter, allowing significant power-saving. They were turned on before spring when sap flow activity increased again.

Chlorophyll fluorescence was measured using a PAM 2100 (Heinz Walz GmbH, Effeltrich, Germany) fitted with DLC-8 leaf clips, which permitted dark acclimation. Three measurements were made per branch, on different needles. The needles were dark-adapted for at least 30 min before measurement. Four parameters were monitored: (a) maximum yield of PSII (Fv/Fm), (b) quantum yield of PSII (ΦPSII), (c) photochemical quenching (qP) and (d) non-photochemical quenching (NPQ) calculated according to [96]. Short-term effects were determined with supplementary measurement close to prescribed burning, 11 (before burning), 12, 13 and 16 June 2014.

Gas exchange was measured using an LI-6400XT portable photosynthesis system fitted with a 6400-01 $CO_2$ injector and a 6400-22L lighted conifer Chamber (LI-COR Inc., Lincoln, NE, USA). To estimate the maximum Rubisco carboxylation rate ($V_{cmax}$), the maximum electron transport rate ($J_{max}$), the rate of triose phosphate export from the chloroplast (TPU) and the mesophyll conductance ($g_m$), A–$C_i$ curves were fitted with 13 steps of $CO_2$: 350, 275, 200, 125, 50, 450, 550, 700, 900, 1100, 1300, 1600 and 2000 $\mu mol \, mol^{-1}$. Other parameters were kept constant (PAR = 1200 $\mu mol \, m^{-2} \, s^{-1}$ with red-blue source (ratio: 0.94–0.06), temperature = 25 °C, air flux = 350 $\mu mol \, s^{-1}$ and vapor pressure deficit (VPD) = 1.2 ± 0.05 kPa). Three readings were made with 30 s intervals between them for each $CO_2$ step. Based on previous works [97,98], we chose to estimate $g_m$, in order to avoid the underestimation of $V_{cmax}$ and $J_{max}$. Parameters were estimated according to

the Farquhar, von Caemmerer and Berry model [99,100]. Estimates were made through the online service presented in [101]. Net photosynthesis (A) and stomatal conductance ($g_s$) were recorded during the first $CO_2$ step (350 μmol mol$^{-1}$) because it is the closest to ambient $CO_2$ concentration. Due to the long time period necessary for A–$C_i$ curves (2.5 to 3 h), the short-term effect of prescribed burning (at the same time as chlorophyll fluorescence measurement) could not be identified.

### 2.5. Data Analysis

Statistical analyses were made using R 3.3.1 for Linux. Differences between the measurements of daily sap flow, gas exchange and chlorophyll fluorescence parameters were tested using a two-way ANOVA and Tukey's range test, when normality and homoscedasticity were verified (by Shapiro–Wilk and Levene's tests). Significant differences were determined at $p \leq 0.05$. The relationship between the sap flow of burned and control trees was tested using Spearman's rank correlation coefficient, because it was non-linear and monotonic. The relationship among the maximum temperature measured under the bark, the temperature measured outside the trunk, the thickness of the bark, and the basal diameter (20 cm above ground) was tested using a general linear model.

## 3. Results

### 3.1. Prescribed Burning Characteristics

The day before the prescribed burn, the mean air temperature and relative humidity were 19.5 °C and 50%, respectively (the last rain event occurred 6 days prior to burning). During the prescribed burn, air temperature and relative humidity ranged from 23.6 to 27.1 °C and 34.3 to 39.5%, respectively.

In order to minimize the fire intensity and resulting trunk damage, the prescribed burning was conducted downslope by the forest management team, following local regulations. The flames were small (length < $0.4 \pm 0.1$ m) and tilted toward the burned area (Figure 2). As a result, no transition from the surface to the crown occurred.

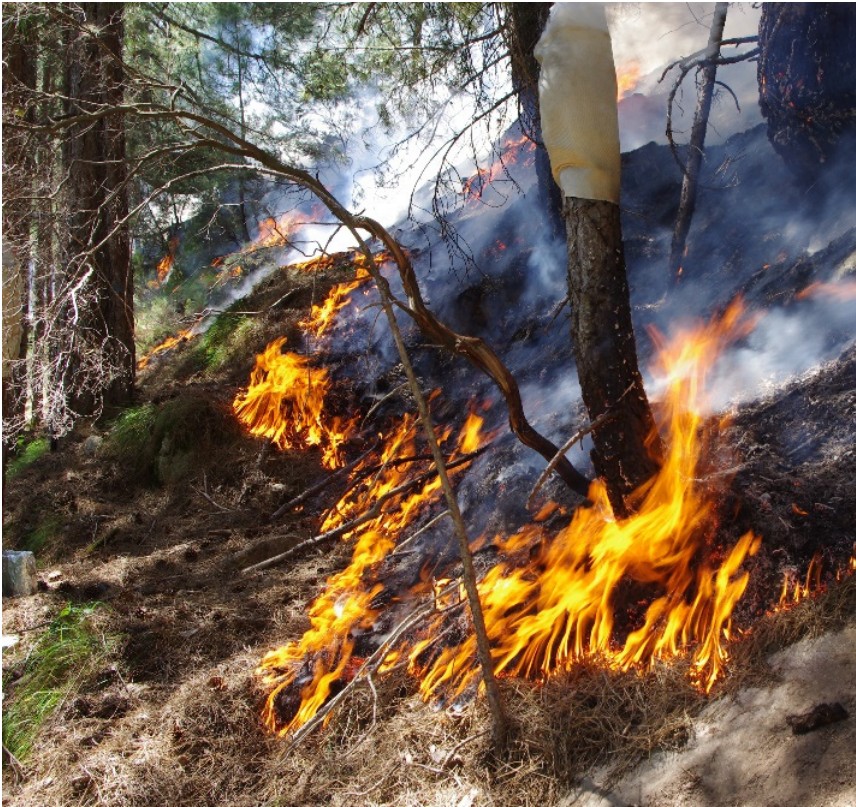

**Figure 2.** The backing flames of the prescribed fire burned pine needle litter and herbaceous strata.

The fire's rate of spread was estimated to be in the range of 0.60–0.84 m/min. Fireline intensity varies in time and space, and the average value was estimated to be between 110 and 160 kW m$^{-1}$, based on surface fuel load and the fire's rate of spread. A typical example of temperature evolution at both the outer and inner bark surfaces is presented in Figure 3.

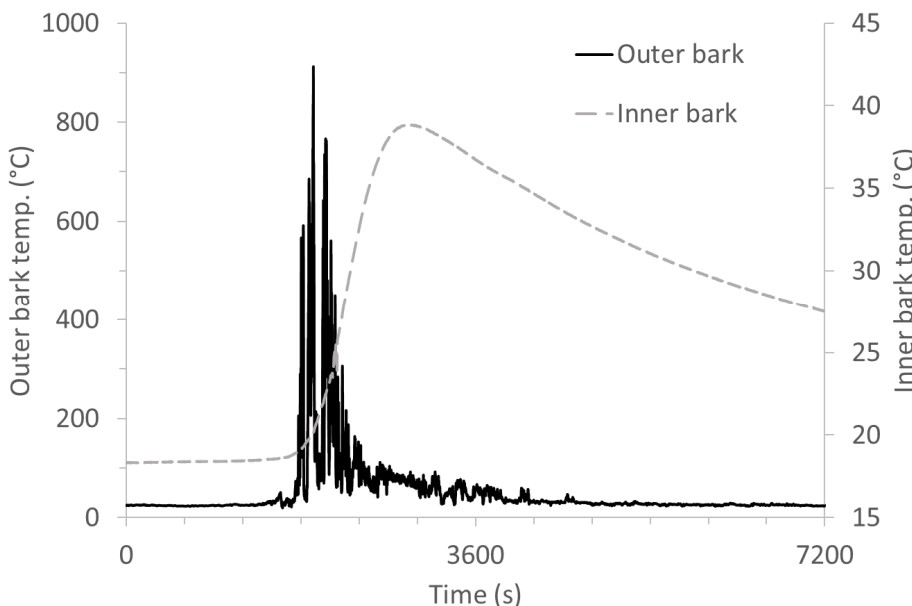

**Figure 3.** Temperatures on the bark surface and inner bark measured during the prescribed burn, for tree B3.

A significant temperature gradient was observed between the outer bark surface and cambial tissues. The maximum temperature recorded at the bark surface and the average flame residence time (duration of exposure to temperatures > 300 °C) were in the range of 845–1278 °C and 220–468 s, respectively. The heat was transferred into the trunk by conduction from the bark surface toward the inner tissues. The resulting maximum temperature measured under the bark was between 18.5 and 56.6 °C. The highest temperature increase (56.6 °C) at the cambium bark interface was measured for tree B5 (Table 4).

**Table 4.** The thickness of the bark, the basal diameter, the maximum temperature measured at the outer (air) and inner (cambium) bark, the time necessary to reach the temperature peak at the level of the cambium and the rate of heating between the outside of the trunk and the cambium, for each tree.

| Tree | Bark Thickness (mm) | Basal Diameter (cm) | Bark Tmax (°C) | | Time to Peak (s) | Heating Rate (°C min$^{-1}$) |
|------|---------------------|---------------------|----------------|--------|------------------|------------------------------|
| | | | Outer | Inner | | |
| B1 | 18 ± 7 | 20.5 | NA | 53.2 | 998 | 4.11 |
| B2 | 13 ± 6 | 17.9 | 999 | 38.6 | 1301 | 1.48 |
| B3 | 14 ± 4 | 14.2 | 911 | 38.9 | 922 | 1.63 |
| B4 | 18 ± 7 | 16.6 | 1278 | 34.7 | 1028 | 1.10 |
| B5 | 20 ± 2 | 17.2 | 955 | 56.6 | 438 | 4.37 |
| B6 | 15 ± 5 | 16.9 | 1029 | 30.1 | 1551 | 0.83 |
| B7 | 26 ± 2 | 21.8 | 1078 | NA | NA | NA |
| B8 | 19 ± 4 | 23.0 | 926 | 24.5 | 1785 | 0.53 |
| B9 | 23 ± 3 | 25.3 | 845 | 26.1 | 1947 | 0.60 |
| B10 | 15 ± 5 | 25.1 | 951 | 18.5 | 2443 | 0.13 |
| B11 | 23 ± 4 | 28.4 | 927 | 27.3 | 1315 | 1.24 |
| B12 | 25 ± 11 | 24.5 | 956 | 28.2 | 1535 | 0.97 |

Differences among trees in the temperature measured under the bark were neither explained by differences in bark thickness, nor basal diameter, nor by the temperature measured in the outer trunk ($p > 0.05$). Furthermore, the temperature also declines slowly after the peak. Due to the low thermal diffusivity of the wood temperature into the trunk, it remained above ambient for a duration of close to 1 h (Figure 2). The char height (the blackening of the tree trunks) ranged from 49 to 120 cm. Thermal pruning of branches and the percentage of scorched needles were both low (<5%), except for one tree (B6) that had ca. 25% of its needles scorched. No visual impact (scars) was observed on the trunk of this tree, and the measured temperatures were similar to those of the other trees.

*3.2. Sap Flow*

Differences in maximum sap flow were observed between burned (up to 1.32 dm$^3$ dm$^2$ h$^{-1}$) and control (up to 0.84 dm$^3$ dm$^2$ h$^{-1}$) trees (Figures 4 and 5), independently of age, height, DBH and position in the stand ($p < 0.05$). Nevertheless, the sap flow of control and burned trees followed similar trends. Both treatments showed strong daily and seasonal variations in sap flow (Spearman's correlation coefficient > 0.7), with higher values being observed during spring and early summer, which then declined sharply at the end of each summer.

Owing to these differences between the trees in the two plots, to compare the pre- and post-prescribed burn flow we standardized the sap flow of control and burned trees. The daily flow (dm$^3$ dm$^{-2}$ d$^{-1}$) of the trees in each plot was divided by its average daily value for the pre-burn period (between 13 July 2013 and 10 Jun 2014). Mean values were 3.372 and 1.976 dm$^3$ dm$^{-2}$ d$^{-1}$ for burned and control trees, respectively. For all measurement periods, normalized daily sap flow values were similar ($p > 0.05$) for burned and control trees. The normalized sap flow is presented by month (average normalized flow of each month for trees in both plots) for the entire study period (Figure 6).

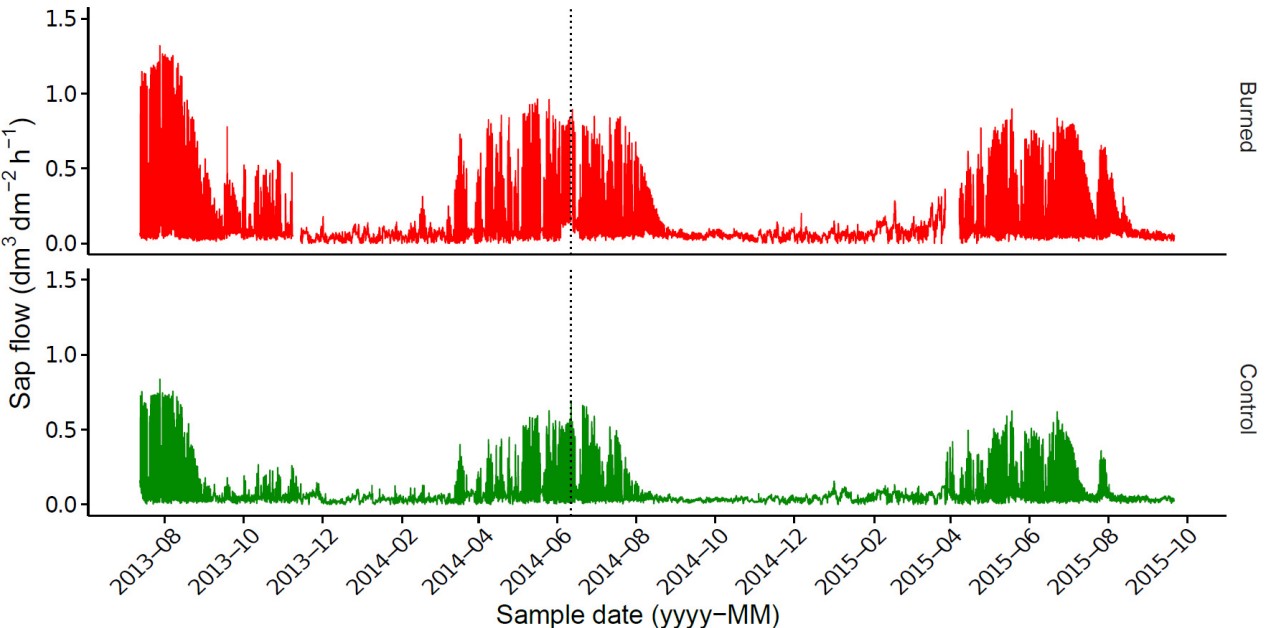

**Figure 4.** Seasonal variation in sap flow from control and burned trees (dm$^3$ dm$^{-2}$ h$^{-1}$) between 13 July 2013 and 26 October 2015. The curves represent the mean value for all 12 trees in each treatment; only one tree by treatment was monitored between 15 November 2013 and 28 April 2014, and between 4 November 2014 and 24 April 2015. The vertical dotted line represents the date of the prescribed burning.

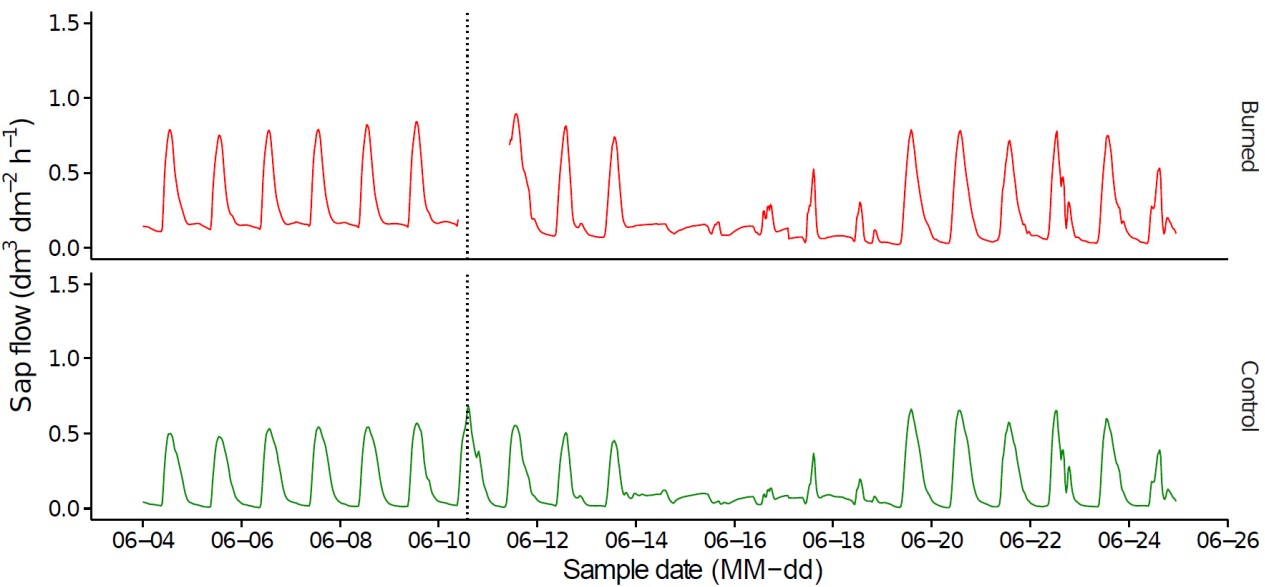

**Figure 5.** Variations in sap flow from control and burned trees (dm³ dm⁻² h⁻¹) between 4 June 2014 and 25 June 2014. The curves represent the average value for all trees in each treatment. The dashed vertical line represents the date of the prescribed burning.

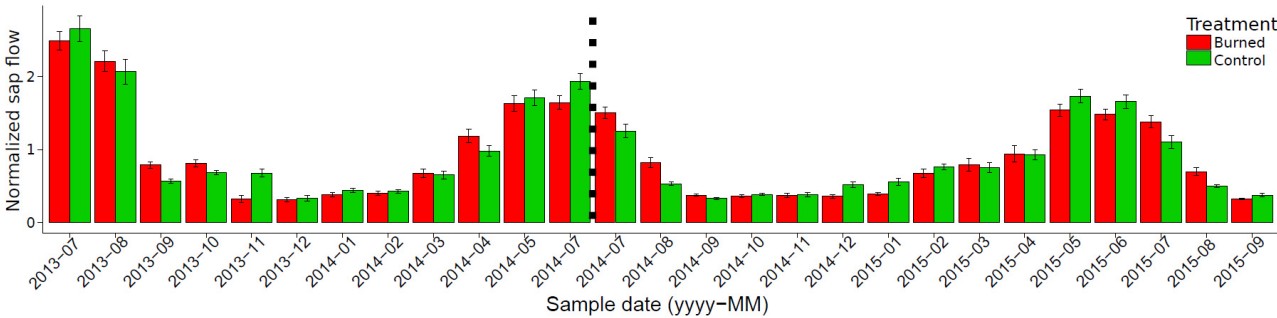

**Figure 6.** Normalized sap flow for control and burned trees. The histograms represent the average value per month of the daily flow, normalized by the average pre-burning value. Error bars represent the standard error. The dashed vertical line represents the prescribed burn.

### 3.3. Chlorophyll Fluorescence

As with the control trees, the values of the chlorophyll fluorescence parameters of burned trees showed a low variation throughout the measurement period (Figure 7; Table 5). Regarding the effects of prescribed burning, small changes in $F_v/F_m$, ΦPSII and NPQ were observed immediately after burning. $F_v/F_m$ and ΦPSII slightly decreased, while NPQ slightly increased. These differences could be attributed to the values observed on a single tree (B6), which caused a variation in the average value of each parameter. This tree had a higher needle scorch rate (≈25%) than the other trees after the prescribed burning. Over the entire measurement period, the values of the chlorophyll fluorescence parameters of burned trees were not significantly different from those of the control trees ($p > 0.05$).

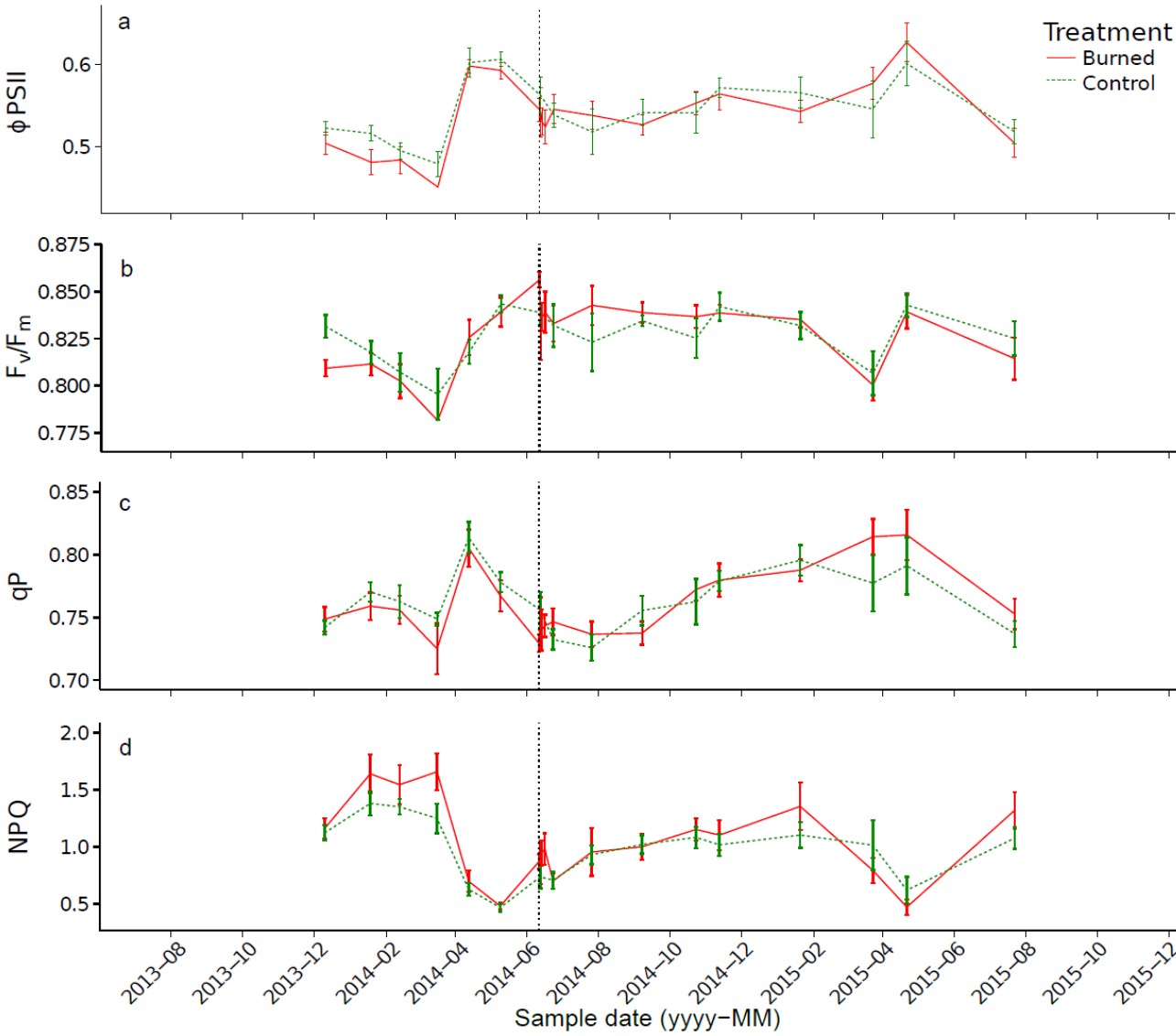

**Figure 7.** Mean values of ΦPSII (**a**), Fv/Fm (**b**), qP (**c**) and NPQ (**d**). Error bars represent the standard error. The vertical dashed line represents the date of the prescribed burning.

**Table 5.** Results of treatment and date effects on chlorophyll fluorescence and gas exchange parameters, by two-way ANOVA (significance level $p = 0.05$).

| Parameters | Treatment | | Date | |
|---|---|---|---|---|
| | *F* Value | *p*-Value | *F* Value | *p*-Value |
| ΦPSII | 0.747 | 0.388 | 10.539 | <0.0001 |
| Fv/Fm | 0.132 | 0.717 | 5.914 | <0.0001 |
| qP | 0.129 | 0.720 | 7.842 | <0.0001 |
| NPQ | 2.791 | 0.096 | 13.291 | <0.0001 |
| A | 7.099 | 0.008 | 22.836 | <0.0001 |
| gs | 15.01 | 0.0001 | 18.19 | <0.0001 |
| gm | 0.002 | 0.968 | 6.126 | <0.0001 |
| Vcmax | 2.541 | 0.112 | 3.597 | <0.0001 |
| Jmax | 2.795 | 0.096 | 10.313 | <0.0001 |
| TPU | 1.294 | 0.256 | 12.530 | <0.0001 |

*3.4. Gas Exchange*

In the control trees, a greater variation in A, $g_s$ and $g_m$ of burned trees was observed, with decreasing values being recorded between July and September 2014 (Figure 8a–c, Table 5).

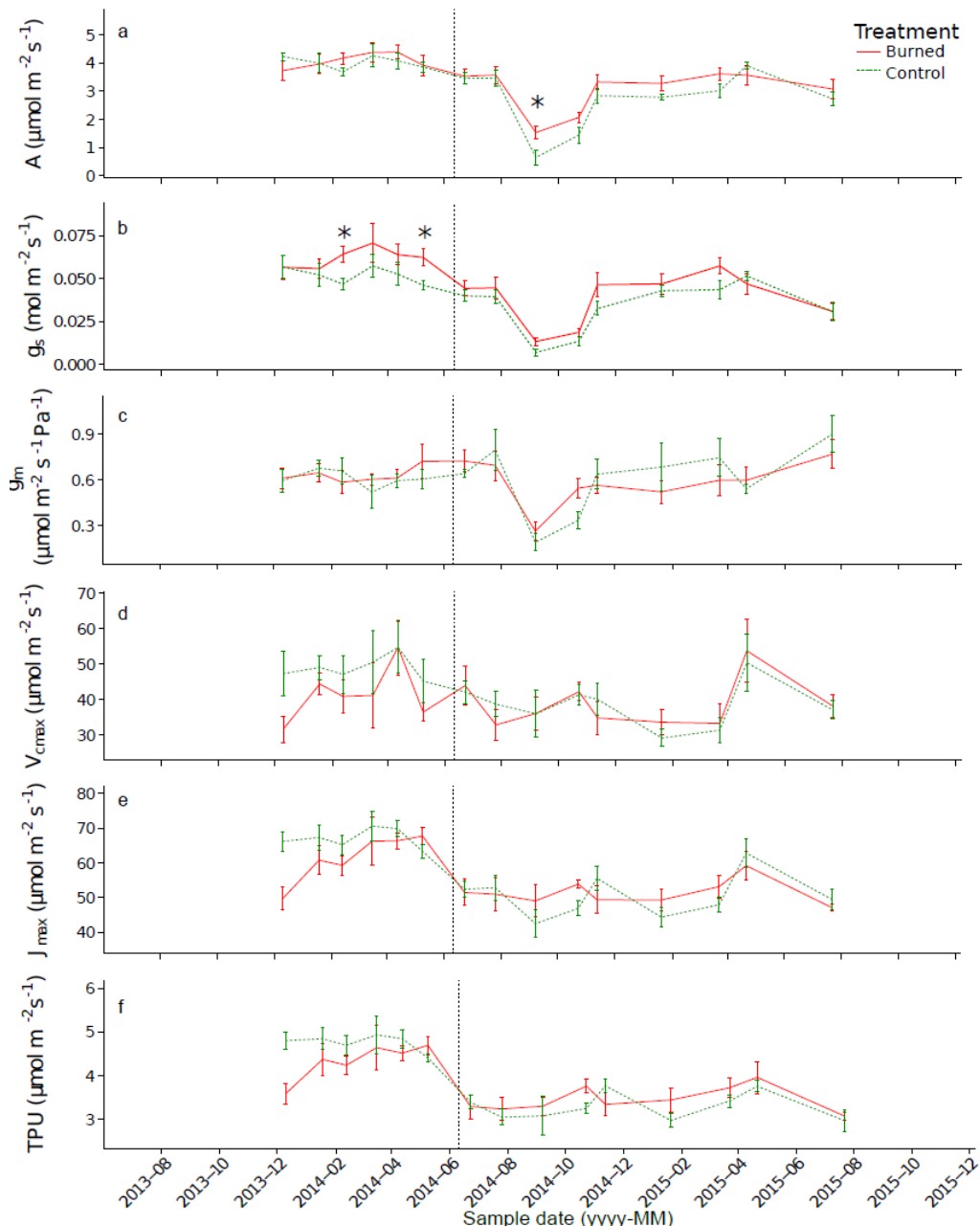

**Figure 8.** Mean values of A ($\mu$mol m$^{-2}$ s$^{-1}$; (**a**), $g_s$ (mol m$^{-2}$ s$^{-1}$; (**b**), gm ($\mu$mol m$^{-2}$ s$^{-1}$; (**c**), Vcmax ($\mu$mol m$^{-2}$ s$^{-1}$; (**d**), Jmax ($\mu$mol m$^{-2}$ s$^{-1}$; (**e**) et TPU ($\mu$mol m$^{-2}$ s$^{-1}$; (**f**). Significant differences between treatments are indicated by asterisks ($p < 0.05$).

$V_{cmax}$, $J_{max}$ and TPU showed low seasonal variation (Figure 8d–f, Table 5). Significant differences between plots were observed for A and $g_s$; A was lower for control trees in September 2014 ($p < 0.05$), while $g_s$ was lower for control trees in February and March 2014 ($p < 0.01$ and $0.05$, respectively). For the other parameters, no significant differences ($p > 0.05$) between control and burned trees were observed.

## 4. Discussion

The prescribed burn was carried out under hard conditions: steep slope, significant surface fuel load, late spring weather. Pre-burn monitoring of control trees showed that June is a period of high metabolic activity, with high sap flow, net photosynthesis and PSII efficiency [102].

During the prescribed burn, ca. 40% of the litter mass was burned, in accordance with previous studies' observations [69,103]. The surface fuel was not fully consumed, with the degradation of only superficial horizons (mostly dry and aerated). Fuel consumption is influenced by different parameters, including litter and live herb loads, fuel moisture, days since the last rain, the season and relative humidity [103]. Moreover, the consumption of branches (of shrubs or on the ground) was not complete, with only the thinnest diameters (<4 mm) playing a role in the dynamics of fire spread [104].

As recommended for prescribed burning practice [9], the intensity of the fire line was low (peaking in the range of 110–160 kW m$^{-1}$). Despite a longer implementation compared to strip head fires [105], the use of a backing fire allowed the limiting of fire intensity. As a result, this was achieved by conducting a downslope fire, resulting in a low rate of spread combined with short flames (<0.4 m). McArthur [106] previously recommended that prescribed burning should be carried out with fire intensities less than 340 kW m$^{-1}$ to ensure that no unacceptable damage occurred to trees. In the present study, the external bark surface of the trees was nevertheless subjected to a rather high heat source, since average peak air temperature and residence time (above 300 °C) were 989 ± 114 °C and 250 ± 115 s, respectively. These results are in agreement with [57] for the same range of litter load. It is worth noting that the residence time measured in the field during the prescribed burning also confirms a radiant heat exposure duration of 300 s as retained by [58] for laboratory tests on bark samples.

As pointed out previously in [58], both the flame residence time surface and maximum temperature of the bark are good predictors of damage to the cambium in field studies where heating is not homogeneous. Nevertheless, despite prolonged flame contact with the base of the trunk during the passage of the fire, the temperature measurements performed under the bark indicate a rather low heat stress to cambial tissues (34.1 ± 11.3 °C). Before the analysis of the physiological indicators (as detailed in the following), these results already suggest a low potential level of damage to the trunk, due to the bark's insulating capability [35,58]. The time needed for the inner tissue temperature to reach a maximum value (time to peak inner temperature) also provides a good indication of the significant protective role played by the bark during a surface fire (Table 4). Due to the low thermal conductivity of bark, the transient increase of the temperature of the underlying cambium occurs at a slow rate. As a result, an average delay of about 23 ± 9 min was observed between the maximum temperatures of outer and inner bark surfaces. The corresponding heating rates at the cambium-bark interface were in the range of 0.13–4.37 °C min$^{-1}$. These low values measured in the field are consistent with previous laboratory findings for the same range of bark thickness [58]. These authors concluded that a bark thickness of 17 mm can be considered as the threshold at which the probability of cambium damage decreases for heat exposure up to 300 s. Our findings confirm that the bark of *Pinus nigra* with 19 ± 4 mm thickness sufficiently insulated the cambium, since the maximum temperatures recorded were below the 60 °C threshold (Table 4). Furthermore, bark has low ignitability, since no ignition was observed during fire spread at the base of the trunk. The variability of the maximum cambium temperature (18.5–56.6 °C) can be attributed to the local surface fuel load at the base of the trees. Nevertheless, the maxima of inner bark temperature are lower than those recorded by [76] under several pine stands (*Pinus nigra* and *Pinus pinaster*) with similar average bark thickness (12.9–19.2 mm). These authors measured higher cambial temperatures in the range of 43.0–151.1 °C during prescribed burnings with a similar fire rate of spread (0.65–0.76 m/min) but a taller flame (0.50–0.62 m). Unfortunately, neither the ground litter load, nor the fireline intensity, nor the flame residence time was provided for

further comparison. Similarly, in [58], the cambium and outer bark temperatures measured in the field for fireline intensities between 11.2 and 32.6 kW m$^{-1}$ were not provided either.

Under-bark temperatures above 40 °C were measured for only two trees of 12 monitored trees (Table 6). Previous authors [76] also observed a small proportion of trees affected by cambial heating. For trees B1 and B5, the temperature remained above 50 °C for 476 s and 423 s, respectively. For the B5 tree, the temperature remained above 55 °C for 248 s. According to [42], this exposure would cause the death of some cells. However, it should be noted that temperature was only measured at a single point (leeward side, where it is theoretically the highest) and the temperature is not constant around the trunk [52,55]. Leeward vortices [51,52,89] and irregularities in the bark can influence heat transfer inside the tree. Nevertheless, the damage to the tree is not only related to the cambium temperature. For instance, for tree B6, the temperature under the bark was not particularly high (30.1 °C) but because of the combustion of shrubs far from the trunk (distance of 2–3 m, shrub height of about 1 m), great crown damage occurred, due to the high release of upward heat by convection to the above needles [89].

**Table 6.** Maximum temperature measured outside the trunk (air) and under the bark and residence time above thresholds. Temperature thresholds were determined from [42].

| | Air | | Under Bark | | | | |
|---|---|---|---|---|---|---|---|
| | Tmax (°C) | Residence Time (s) | Tmax (°C) | Residence Time (s) | | | |
| Tree | | >300 °C (s) | | >55 °C (s) | >50 °C (s) | >45 °C (s) | >40 °C (s) |
| B1 | NA | NA | 53.4 | | 476 | 918 | 1454 |
| B2 | 999 | 416 | 38.6 | | | | |
| B3 | 911 | 371 | 38.9 | | | | |
| B4 | 1278 | 413 | 34.7 | | | | |
| B5 | 955 | 468 | 56.6 | 248 | 423 | 668 | 1220 |
| B6 | 1029 | 333 | 30.1 | | | | |
| B7 | 1078 | 428 | NA | | | | |
| B8 | 926 | 220 | 24.5 | | | | |
| B9 | 845 | 266 | 26.1 | | | | |
| B10 | 951 | 222 | 18.5 | | | | |
| B11 | 927 | 233 | 27.3 | | | | |
| B12 | 956 | 277 | 28.2 | | | | |

Previous studies have shown that cell death is a dependent process that is influenced both by the heating rate and time, i.e., necrosis occurs at a temperature below 60 °C if the exposure time is sufficient [38,89]. A temperature of 45 °C would be lethal for cells with an exposure time of several tens of minutes, while a few tens of seconds would be enough to cause death at 63 °C for *Populus tremuloides* Michx. cambium and phloem cells, in a hot water bath [42]. The authors also observed differences in heat sensitivity among species. For example, the cell death rate is lower for *Pinus contorta* Douglas ex Loudon than for *P. tremuloides*, *Picea engelmannii* Parry ex Engelm and *P. menziesii*. Of note, not all cells die simultaneously. For example, in [42], at 55 °C, the first cell deaths were observed after less than 200 s, while necrosis was complete after more than 600 s. Finally, the death of some cells does not mean that the tissues (including cambium and phloem) are no longer functional [89].

Many studies have focused on meristem necrosis because of the ability of plants to regenerate other tissues and organs that have been damaged. Thermal damage to the trunk might have different impacts depending on the rate of necrosis, but also depending on the affected tissue. If the cambium dies over the entire circumference of the trunk, regeneration and secondary growth in this area are not possible. Of note, necrosis of the cambium by thermal stress is also accompanied by necrosis of the phloem. In this case, the foliage continues to fix carbon, but the sugars produced are not transported by the phloem to the roots. As a result, the activity of the fine roots decreases and the tree eventually dies due to water stress [89]. Necrosis of the cambium localized at one point produces a scar; this

phenomenon is generally observed downwind of these vortices [52,55,89]. This type of injury has little or no effect on crown morphology and the physiological functioning of trees [54,107] but it does deform wood, reducing its commercial value.

Heat stress in the xylem can lead to cavitation. This phenomenon is induced by the surface tension of the sap and varies inversely with temperature [108]. Significant cavitation was observed for branches subjected to temperatures of 65 °C and 95 °C for 5 min (in a hot water bath [43]). Due to its thickness and its position (inner to the cambium), xylem is difficult to embolize in its entirety [38,47]. In our experiment, the temperature under the bark, at the outer limit of the xylem, was much lower. Recovery of xylem conductivity is possible through the formation of new vessels [109] this implies that the cambium is not necrotized.

Prescribed burning has different and opposing effects on sap flow, depending on the intensity, season and tolerance of the species. Renninger et al. [69] observed a decrease in sap flow immediately after burning and explained it based on a decrease in leaf area and, probably, on thermal damage to the cambium and phloem. In comparison, one month later, the authors observed an increase in sap flow, which was explained by an increase in the total leaf area of pines through the development of many epicormic buds. In [69], burning was particularly severe, with average fuel consumption at ground level of about 327 g m$^{-2}$ and stem combustion of 363 g m$^{-2}$. While in our study, despite a surface fuel consumption in a close range (270 g m$^{-2}$), branch combustion was very low. This is due to the natural lack of low branches in such old and dense stands.

Previous authors [54] found that burned trees (70–90% of the circumference), in particular, had a similar flux to control trees. These results demonstrate the high tolerance of the trees (*Pinus halepensis* Mill. in this study) to warming up, despite the temperature and residence time achieved over a large part of the cambium. In comparison, the sap flow of trees that completely burned decreased sharply the week after treatment. These results might be explained by the absence of xylem ring formation in the year of treatment, indicating damage to the phloem, which impairs proper root functioning, or it might be based on the phenomenon of xylem cavitation.

The sap flow of trees from both plots followed the same seasonal and daily pattern (Figures 3 and 4). These results were observed for both the pre-burning and the post-burning periods. One week after the prescribed burning, standardized sap flow was slightly greater for control trees (Figures 6 and 7). However, no significant differences ($p > 0.05$) were observed between the two plots. Few visible burning effects on needles were observed. Thermal cutting and the percentage of scorched needles were both low and, therefore, leaf area reduction was low and probably insufficient to induce a significant decrease in sap flow.

A previous study [68] highlighted damage to the PSII reaction center (decreased $F_v/F_m$ and/or ΦPSII) a few months (three to 12 months) after prescribed burning on *P. halepensis* and *P. laricio*. This damage was explained by thermal stress, whereby the critical temperature for activity of the PSII is believed to be between 45 °C and 50 °C for Mediterranean [110] and temperate tree species [111]. Immediately after prescribed burning, slight decreases in ΦPSII and $F_v/F_m$ and a slight increase in NPQ were observed (Figure 7), but these differences were not significant ($p > 0.05$). No differences were observed between the control and burned trees and between pre- and post-prescribed burning. The lack of a significant effect of prescribed burning on the efficiency of the PSII might be explained by the height of the sampled branches (5–6 m). Indeed, temperature decreases sharply with increasing height. Cannac et al. [75] measured temperature at different heights above the ground surface, with soil litter masses similar to those of our experiment. The authors found that the temperature was about 60 °C at 3 m in height. Extrapolating these results using non-linear least squares regression, the temperature at 5 and 6 m is expected to be about 40 °C and 33 °C, respectively (Figure 9).

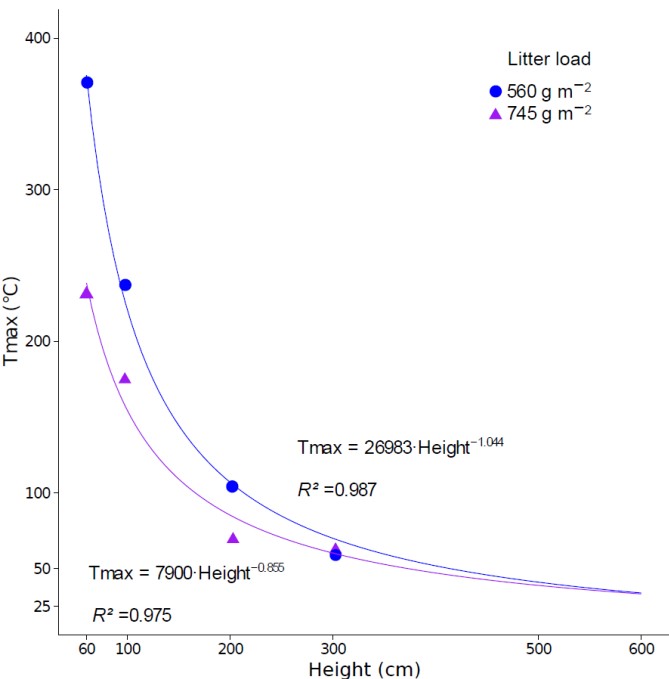

**Figure 9.** Maximum temperature measured at different heights during two prescribed burns, with litter loads of 560 and 745 g m$^{-2}$ (the average fuel load was 656 g m$^{-2}$ in present study), according to [75] and temperature extrapolation up to 600 cm high using a non-linear least squares model.

Trunk heating (on *P. halepensis*) negatively affects photosynthesis for totally burned trees, compared to the control and partially burned trees [54]. This decrease in A was associated with a decrease in $g_s$ and sap flow, probably due to cavitation and/or tissue destruction of the phloem. Cannac et al. [62] observed a decrease in the concentration of pigments (chlorophyll a and b and carotenoids) over a short time (one month) following prescribed burning on *P. laricio*, collected at 2 m and 6 m height. Because the temperature is low at 6 m, the direct effect of thermal stress on needles was, therefore, unlikely in our experiment.

For gas exchange parameters, only A showed a significant difference between the trees of the control and burned plots after the prescribed burning (Figure 8). The value of A was higher in September 2014 for burned trees. The absence of negative effects on gas exchange parameters might be explained by the important insulating role played by the bark and the height of the sampled branches. The difference in A observed between the burned and control trees in September 2014 might be explained by an increase in available nitrogen, due to the mineralization of soil organic matter. Indeed, fire causes an immediate decrease in the concentration of organic nitrogen by volatilization, but a substantial part is converted to inorganic forms ($NH_4^+$ and $NO_3^-$) that are available to plants [112]. According to literature, a nitrogen supply could lead to an increase of $V_{cmax}$ (*Pinus rigida* Mill. [69]) or an increase in A and $g_s$ (for different species of angiosperms [59]) a few months after burning.

At the end of the summer following the prescribed burning, sap flow, $g_s$, $g_m$ and A values of trees decreased significantly in both plots. The similarity between the results for burned and control trees indicates that these decreases occurred due to the summer drought and the water-saving strategies of this species [102,113]. In addition, [114] suggested that a given species displays fixed plant functional traits, even during extended droughts and in post-fire regeneration. In the control trees, the decrease in A values in burned trees was probably caused by stomatal closure (reduction of $g_s$). At the same time, the chlorophyll fluorescence parameters remained stable. Prescribed burning likely does not negatively affect the physiology of trees or their resistance to summer drought.

## 5. Conclusions

The present study provides new insights on the effects of fire on the vitality of *P. laricio* during prescribed burning, particularly with respect to heat transport through the bark, from the outside to the inside of the tree. The rate of heat transfer from the bark surface to the cambium was low, confirming the efficient insulating role of the bark of *P. laricio* during a low-intensity surface fire. The observed temperatures and residence times were, in most cases, below the damage threshold reported in the published literature. The reality of field experiments exhibited a noticeable variability of the temperature measurements. However, both short- and long-term effects on the trees were more homogeneous. Our findings demonstrated that prescribed burning, with a fire intensity of up to 160 kW m$^{-1}$, had no adverse effects on the physiology of 2-cm bark thickness *P. laricio* trees, including during the summer drought period.

For the sustainable management of forests, especially in natural regeneration stands, it is essential to understand the damage threshold and the effects of prescribed fires on different age classes. Future works will focus on a larger range of tree diameters, including young specimens.

The consequences of prescribed burning, particularly in the spring, are still highly debated. These results show that a good compromise can be found to significantly reduce wildfire risk while exposing the trees to acceptable temperature ranges, whatever the season.

**Author Contributions:** Conceptualization, F.M. and L.F.; methodology, F.M. and L.F.; instrumentation, F.M., L.F. and G.L.; data collection, F.M. and G.L.; data analysis, F.M., L.F. and G.L.; writing F.M., L.F. and G.L.; funding acquisition, L.F. All authors have read and agreed to the published version of the manuscript.

**Funding:** This research was funded by the Collectivité de Corse and Office de l'Environnement de la Corse.

**Conflicts of Interest:** The authors declare no conflict of interest. The funders had no role in the design of the study; in the collection, analyses, or interpretation of data; in the writing of the manuscript, or in the decision to publish the results.

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
