# Peer review of "Influence of Prescribed Burning on a Pinus nigra subsp. Laricio Forest: Heat Transfer and Tree Vitality"

_forests, doi:10.3390/f12070915_

Round 1

Reviewer 1 Report

The writing is very clear, neat and appropriate for a scientific article. Few ambiguities (or apparent inaccuracies) are detected and I have pointed them out and they will be easy to correct. I have enjoyed reading it. Glad to see results and conclusions in this research. All in all, it's a very good job.

Author Response

We first thank the reviewer for such detailed comments which helped us to significantly enhance our present and future works. We included our responses in the pdf file.

Reviewer 2 Report

The authors presented an excellent manuscript with a comprehensive review of the literature and an effective design. While I struggled to find fault with their design or presentation, my main criticisms are their ignoring the role of fire as an essential ecological process in addition to being a threat from a human perspective and the relatively narrow range of tree diameters chosen for the study. The threat of wildfire (assuming the authors mean high intensity, as the term wildfire generally refers to ignition and management intent, not intensity) is only a small facet of the role of fire. An analogy would be to try and suppress all rainfall because floods are damaging.

The authors focused on the impacts of fire on mature tree stems and their physiology and I really could find no fault with their approach and execution. However, the choice of only mature trees should be discussed as well as how smaller diameter saplings would be affected. This doesn't negate their findings but need more discussion. Also, how would higher intensity fires affect these trees? Can the authors infer at what point a damage threshold would be crossed? I applaud their careful work which made the review process easy.

Author Response

We first thank the reviewer for his comments which helped us to significantly enhance our present and future works. We have adressed a point by point response in the .doc file.

Reviewer 3 Report

Influence of a prescribed burning on a Pinus nigra subsp. laricio forest: heat transfer and tree vitality

Main topic: Authors measured heating of outer bark and cambium of Pinus nigra subsp. laricio during a spring prescribed burn and then conducted physiological monitoring of the trees for 2.5 years to determine longer term effects of the prescribed burn. Measurements included sap flow, chlorophyll fluorescence, and gas exchange measurements. They found that bark insulated cambium sufficiently to prevent heat related damage or stress. Physiological measurements showed the fire had no long-term negative effects on trees in the burn unit.

Importance: This study was unique in that the measurements were taken in the field during an actual prescribed burn versus in a lab.

General comments:

Twelve trees seem like a relatively small sample size but given the intensity of the physiological measurements this may have been necessary.

Writing style would benefit from editing by a native English speaker to catch small nuances of the language and make the writing clearer and more concise.

Several words are used to talk about the “under bark” or “underneath tissue”. I would change those to “inner bark” or “cambium” or “cambial tissue” whichever you think is more accurate and use that term consistently throughout the paper.

What is the fire history of the area? It seems that Pinus nigra subsp. laricio, like most pines, is fire dependent to some extent but that due to fire suppression fuels have built up so the fire regime is changing from a frequent surface fire regime to a stand replacement regime. The objective of the Rx-burns are to bring the system back to its historical fire regime (and prevent catastrophic wildfires), but there is some public resistance to burning (especially in the spring). If this is the case, it would be good to explain that a little more clearly in the introduction.

Line comments

2 - italicize genus and species

8 – facilitate fire mitigation – Does that mean prevent catastrophic crown fires?

10 – Delete: “Literature review exhibited that; Delete ending s in transfers

13 - Is Pinus nigra subsp. laricio a fire dependent species or a fire sensitive species? Is the forest solely comprised of Pinus nigra subsp. laricio? Does it have a common name?

14 - change in to of; what does “some herbaceous” mean? Some live herbaceous plants? Some live grasses and forbs? Pine litter depth would be nice to know as well as height of the herbaceous fuel.

15 - 12 trees seems like a very small number of trees. What was the acreage of the burn? What season did the burn happen? Were trees just measured on one Rx-burn?

18 – comma after C, and then add an a before critical. 60 ⁰C, a critical threshold

23 – Remove “Thanks to” and “good”. The thermal insulation capability of the bark…

23-24 – underneath tissue = cambium?

32-33 – burned surfaces = hectares burned

34 – climate warming = global climate change

35 – What does “increase fire regime” mean? Increase fire severity? Fire season duration? Fire intensity? Fire behavior?

37 – Are these forests fire sensitive or fire dependent? Many pines are adapted to fire, but with the build-up of fuels the fire regime has changed from a surface fire to catastrophic crown fire. Is that the case here? A little more background information about the fire history of the area would be helpful.

38-40 – run-on sentence

45 – change “hardly reachable” to “difficult to reach”

46-48 The removal of these ladder fuels will reduce the potential of fire transition from the surface to crown and create more spatial variability in the structure of the forest.

53-55 –But Rx-fires often produce less particulate matter than wildfires and are known to increase biodiversity. See references:

Prunicki, Mary et. al. 2019. The Impact of prescribed fire versus wildfire on the immune and cardiovascular systems of children. European Journal of Allergy and Clinical Immunology.

Vander Yacht, A.L., Keyser, P.D., Barrioz, S.A. et al. Litter to glitter: promoting herbaceous groundcover and diversity in mid-southern USA oak forests using canopy disturbance and fire. fire ecol 16, 17 (2020). https://doi.org/10.1186/s42408-020-00072-2

58 – Is Table 1 necessary as you cite many (if not all) of those papers in the following paragraph?

61 – This paragraph is pretty long.

61-63 Other suggested study developed mortality models based on char height and scorch:

Keyser, T.L., McDaniel, V.L., Klein, R.N., Drees, D.G., Burton, J.A., Forder, M.M. 2018. Short-term stem mortality of 10 deciduous broadleaved species following prescribed burning in upland forests of the Southern US. International Journal of Wildland Fire. International Journal of Wildland Fire 27(1): 42-51. [http://www.publish.csiro.au/wf/WF17058]

Keyser, T.L., McDaniel, V.L., Klein, R.N., Drees, D.G., Burton, J.A., Forder, M.M. 2017. Validation and development of postfire mortality models for upland forest tree species in the southeastern United States. Pages 98-109 In: Keyser, C.E., Keyser, T.L., eds. Proceedings of the Forest Vegetation Simulator e-Conference. E-Gen. Tech. Rep. SRS-GTR-224. Asheville, NC: U.S. Department of Agriculture, Forest Service, Southern Research Station. 12 p.

71 – Begin new paragraph with: “The rate of heat transmission….”

84-85 – Studying fire under field conditions is very important to really understanding its effects.

81 – Begin new paragraph after “However…”

87 – add s to plant; change as to like

102 – Begin new paragraph with “Combined…”

141 – delete blank line

142 – italicize genus and species; why is “var. corsicana” here when it is nowhere else in the paper? If this is the full name of the species mention that you will be shortening the name to Pinus nigra subsp. laricio for the rest of the paper. Can we get a common name for this species?

162 – what does “exploited” mean? Burned? Thinned? Cut?

167 – change “exposition” to “aspect”

175 – write out what VPD is – Vapor Pressure Deficit (VPD)

186 and 189 – Might be nice to have a column for average DBH, Height, Age and TDP for easy comparison of trees in control and burn treatments.

192-193 – Rewrote a couple sentences: The upper slope of the burn unit was lit by hand using fire torches. (drip torches?). This intentionally created a backing fire with very low fire intensity.

193-194 – Were trees far enough interior of the burn to not be affected by these fire controls?

199 - change at to on

200 - since fire intensity is greatest in this area due to flame vortices

202 – should “trunk” be “bark”?

203 – change “from” to “to”, phloem is part of the cambium. Maybe just say cambium and delete phloem. I saw this a few times in the paper so will need to change other places if you see fit.

213 – change “on” to “into”

215 – change “burning” to “burn”

224 – Did you get a litter depth?

237 – Change “Five to six meters height sun-exposed branches” to “Sun-exposed branches approximately five to six meter high” were sampled…

238 – run on sentence: change ”position, they” to “position. They”

249 – change “exposition” to “aspect”

255 – add an “and” before battery or make plural if more than one battery was used (batteries).

299 – change “prescribed burning” to “the prescribed burn”

301 - change “prescribed burning” to “the prescribed burn”; change “were in the range of” to “ranged from”

303- 304 - Suggested rewording: Fire behavior was low with flame lengths of the backing fire ranging from …..

306 – Suggested rewording: Backing flames of the prescribed fire burned through pine needle litter and herbaceous plants.

312 – change “under bark” to “inner bark” or “cambium” - need to change in the figure too; change “burning” to “burn”

315 – change “underneath tissues” to “inner bark” or “cambial tissues” or “cambium”

326 – I think you are referring to char here. Perhaps say “Char height (the blackening of the tree trunks) ranged from 49 to 120 cm.”

355 – change “pre-burning” to “pre-burn”

363 – change “prescribed burning” to “prescribed burn”

377 – change “burning” to “burn”

381 – spacing off in third column, second row

397 – change “burning” to “burn”

401 – change “During prescribed burning” to “During the prescribed burn”

405 – remove “air”

407 – change “being involved with the spread dynamics of the fire” to “being consumed.”

422 – change “to the” to “of”

426 -427 – confusing sentence.

428 – change “thanks” to “due”

431 - change “Thanks” to “Due”

434 – change “measured at bark outer and inner surfaces.” to “of outer and inner bark surfaces.”

440 – change “played an efficient insulating role” to “ sufficiently insulated cambium”

443 – move cambium between maximum temperature (maximum cambium temperature)

446-447 – Delete “In this previous study, the authors” and just state: “Espinosa et al. (2018) measured…”

457 – Delete “the results of”

459 – Delete “on the other hand, the”

460-461 – put period after references; Delete “in particular due to” and capitalize Leeward

462 Delete “that”

472-473 – Isn’t phloem part of the cambium?

479 – Isn’t phloem part of the cambium?

508 – change “opposite” to “opposing”

518 – Do fire fighters cut lower branches and remove shrubs?

589 – I would something like “from the bark surface to the cambium”

594 - tree should be plural; run on: “homogeneous. Our findings”

Author Response

(The authors gave the same response as above.)
